# Investigating the Impact of the Number of Medication Use on Depression Among Hypertensive Patients: Results from the National Health and Nutrition Examination Survey Database

**DOI:** 10.3390/medicina60101708

**Published:** 2024-10-18

**Authors:** Fahad T. Alsulami, Atiah H. Almalki, Majed A. Algarni, Mohammad S. Alzahrani, Yousef Saeed Alqarni

**Affiliations:** 1Department of Clinical Pharmacy, College of Pharmacy, Taif University, P.O. Box 11099, Taif 21944, Saudi Arabia; m.alqarni@tu.edu.sa (M.A.A.); m.s.alzahrani@tu.edu.sa (M.S.A.); 2Department of Pharmaceutical Chemistry, College of Pharmacy, Taif University, P.O. Box 11099, Taif 21944, Saudi Arabia; ahalmalki@tu.edu.sa; 3Addiction and Neuroscience Research Unit, Health Science Campus, Taif University, P.O. Box 11099, Taif 21944, Saudi Arabia; 4Department of Pharmacy Practice, College of Clinical Pharmacy, Imam Abdulrahman Bin Faisal University, P.O. Box 1982, Dammam 31441, Saudi Arabia; ysalqarni@iau.edu.sa

**Keywords:** polypharmacy, hypertension, depression, cardiovascular diseases, mental health

## Abstract

*Background and Objective:* Hypertension is a prevalent chronic condition often treated with multiple medications, leading to polypharmacy, which can heighten the risk of adverse drug reactions and contribute to psychological issues like depression. This study aimed to investigate the relationship between polypharmacy and depressive symptoms in hypertensive patients using data from the National Health and Nutrition Examination Survey (NHANES) from 2017 to 2020. *Materials and Methods*: This study utilized data from the National Health and Nutrition Examination Survey (NHANES) collected between 2017 and March 2020. *Results*: Among 2543 hypertensive participants, 12.3% met the criteria for depression. The findings revealed that patients using 11 or more medications were ten times more likely to experience depressive symptoms compared to those taking 1 to 2 medications (OR = 10.06, *p* < 0.001). Additionally, younger age (18 to 45 years), female gender, and lower educational attainment were significantly associated with higher rates of depressive symptoms. Specifically, females were 1.47 times more likely to experience depression compared to males (*p* = 0.032). *Conclusions*: This research highlights the substantial impact of medication burden on mental health among hypertensive patients, emphasizing the need for tailored clinical interventions for this vulnerable population.

## 1. Introduction

Hypertension is a long-term medical condition characterized by persistently elevated arterial blood pressure and is a leading contributor to cardiovascular diseases (CVDs), stroke, peripheral artery disease, vision impairment, and kidney failure, and is experienced by millions of people [1]. When treating this condition, there is a tendency to prescribe multiple drugs, which is why treating hypertension and the associated complications always involves polypharmacy [2]. Although polypharmacy may be useful clinically, it has been identified as a double-edged sword, as the cumulative pharmacological burden heightens the risk of adverse drug reactions (ADRs), including psychological effects such as depression. Depression is one of the most common mental disorders and it has been recognized that it commonly coexists with chronic diseases, which worsens the effectiveness of treatment and reduces patients’ quality of life [3].

Polypharmacy and depression are two factors that have been discussed in the literature with significant interest, especially with regard to elderly patients who require multiple medications for different diseases [4]. A systematic review by Masnoon et al. [5] pointed at polypharmacy as a major cause of ADRs which includes psychological disturbance and depressive symptoms. Managing many medications can be stressful for the physical and mental health of patients with hypertension on top of the existing emotional and mental health issues [6]. Polypharmacy also leads to a high probability of drug interaction and causes what may be called medication burden, a factor that makes the patient psychologically stressed due to the complexity of managing the medications [7].

In one study, the interconnection between hypertension, depression, and adherence to blood pressure-reducing medications was explored by researchers. Data from approximately 852 individuals with chronic hypertension who developed depression over nine years were analyzed and compared with 2359 corresponding hypertensive controls by the researchers. The findings demonstrated that in men who had both hypertension and depression, the rate of non-adherence to medication increased by a factor of 1.52 in the years following the onset of depression compared to the rate of non-adherence observed during the period prior to the development of depression [8].

Besides polypharmacy, some other factors like gender and age have also been found to worsen depressive symptoms in chronic diseases. Biological, social, and emotional stressors were identified by Paramba et al. [9] to have contributed to the higher rates of depression among women, young adults, and patients with chronic illnesses than male and elderly patients [9]. For example, women might be influenced by hormonal changes, while young people could experience psychological challenges related to managing a chronic illness, in addition to the potential effects of taking multiple medications simultaneously [10].

In another study, a long-term cohort analysis was conducted by researchers involving patients aged 20 to 70 years who commenced antihypertensive treatment with the aim of investigating the impact of symptoms of depression and anxiety on their adherence to medication. Throughout a one-year period, 178 patients were tracked, and various questionnaires were employed to evaluate their psychological well-being, quality of life, and the severity of depressive symptoms. It was found that adherence to the prescribed medication regimen was not achieved by 52.6% of these patients, with adherence defined as taking less than 80% of the prescribed pills. The research further revealed that patients who exhibited mild depression, as assessed using a standardized scale, were 2.48 times more likely to be non-adherent to their medication regimen in the subsequent three months. Similarly, patients who exhibited mild anxiety on the scale had a 1.59-fold higher likelihood of non-adherence to their medication [11]. These relationships remained statistically significant even after adjusting for other potential risk factors. Further analysis, which accounted for various variables, demonstrated that individuals with mild depression and anxiety were significantly more prone to discontinuing medication adherence within the following three months. Specifically, those with mild depression were 2.48 times more likely to become non-adherent, while those with mild anxiety were 1.59 times more likely to exhibit non-adherence. Based on these findings, the authors concluded that screening for symptoms of depression and anxiety is crucial for identifying patients who are at risk of failing to adhere to their antihypertensive medication [11].

Although there is increasing evidence concerning the association between polypharmacy and depression, most of the conducted studies involved elderly patients or patients with chronic diseases [12]. The relationship between polypharmacy and depression, more specifically in hypertensive patients, is still quite unknown, even though hypertensive patients are at high risk of both physical and psychological complications. Since hypertension is known to be related to poor mental health in patients [13], it becomes important to know the degree to which polypharmacy would worsen depressive symptoms among the patients. Furthermore, it is valuable to identify how sociodemographic factors moderate the relationship between medication use and depression because such information will help clinicians develop individualized interventions.

The present study aims to plug this gap by examining the relationship between polypharmacy defined as the number of medications taken and depressive symptoms among hypertensive patients in a large nationally representative sample [14]. Drawing on the National Health and Nutrition Examination Survey (NHANES) for the period of 2017 to 2020, we seek to establish whether polypharmacy is independently associated with depression while adjusting for age, gender, marital status, and education level [15]. Thus, in addition to replicating extant research on the psychological consequences of polypharmacy, this paper enriches the literature by analyzing the effects of polypharmacy specifically among hypertensive patients—one of the most vulnerable groups in terms of both physical and psychological well-being.

## 2. Materials and Methods

The National Health and Nutrition Examination Survey (NHANES) is a comprehensive national study that evaluates the health and dietary status of the non-institutionalized civilian population across the United States [16]. Conducted biennially by the National Center for Health Statistics (NCHS), a division of the Centers for Disease Control and Prevention, this ongoing program began in 1999. It employs a stratified, multistage probability sampling method, encompassing around 5000 hypertensive participants nationwide in each cycle [17]. The NHANES protocol received approval from the NCHS Research Ethics Review Board, and all hypertensive participants provided written informed consent prior to their involvement in the study [18]. Since NHANES data are publicly accessible and available for free, obtaining approval from a separate medical ethics committee was not required.

This study utilized data from the National Health and Nutrition Examination Survey (NHANES) collected between 2017 and March 2020. The inclusion criteria were as follows: (1) hypertensive participants had to be interviewed and assessed; (2) they needed to be 18 years of age or older; and (3) they had to have a diagnosis of hypertension. Exclusion criteria applied to individuals who did not provide information regarding their number of medications used or who had insufficient data on depression. Hypertensive patients were identified based on whether participants had been informed that they had high blood pressure (BPQ020) and whether they were taking prescription medication for hypertension (BPQ040A).

### 2.1. Assessment of Depression

The assessment of participants’ depression was conducted at a mobile examination center by trained interviewers utilizing a computer-assisted personal interviewing system. Depression was measured using the Patient Health Questionnaire (PHQ-9) score. The PHQ-9 is a tool designed to screen for depression, comprising nine questions that assess the occurrence of depressive symptoms over the previous two weeks [18]. Responses are categorized as “not at all”, “several days”, “more than half the days”, and “nearly every day”, with corresponding scores of 0, 1, 2, and 3. The total score can vary from 0 to 27, with a score of 10 or above indicating the presence of depression. This threshold is widely recognized and validated for use in primary care environments [18,19].

A meta-analysis of individual participant data evaluated the PHQ-9 scores against a validated diagnostic interview for diagnosing major depression. The findings revealed that the PHQ-9 demonstrated greater sensitivity compared to the semi-structured diagnostic interview, which was primarily designed for clinical management [20]. A cut-off score of 10 or higher was determined to optimize both sensitivity and specificity across the entire sample and various subgroups [21]. Furthermore, another study highlighted that when mental health professionals’ interview results were used to assess the effectiveness of PHQ-9, the sensitivity and specificity for detecting major depression with a PHQ-9 score of 10 or more could reach up to 88% [22].

### 2.2. Assessment of the Number of Medication Use

The independent variable is the number of medications used and this variable is operationalized by RXDCOUNT. This variable covers the total number of prescribed drugs that the hypertensive participants were using at the time of the survey. Number of medications refers to all medications used by hypertensive patients, not just antihypertensive drugs. Based on the number of medications taken, hypertensive participants were categorized into six distinct groups: those using 1 to 2 medications, 3 to 4 medications, 5 to 6 medications, 7 to 8 medications, 9 to 10 medications, and those using 11 or more medications.

### 2.3. Assessment of Covariates

Sociodemographic factors included age (18 to 45 years old, 46 to 65 years, and 66 to 80 years old), gender (male/female), marital status (never married, divorced/separated/widowed, married, or living with a partner), race (non-Hispanic white, Mexican American, non-Hispanic black, and others), country of birth (U.S. country of birth and non-U.S. country of birth), and education (high school or less, college degree graduate degree).

### 2.4. Statistical Analysis

Due to the complex multistage sampling design of NHANES, the data were combined and weighted according to in accordance with NHANES guidelines. Hypertensive participants were initially categorized based on the presence or absence of depression. Then, the baseline characteristics of the hypertensive participants were compared utilizing the weighted chi-square test for categorical variables. For descriptive analysis, the variables were reported as weighted means accompanied by their standard deviations. Categorical variables were represented as weighted counts and corresponding percentages. Subsequently, weighted multivariate logistic regression models were employed to investigate the association between depression and the number of medications used. The outcomes of the weighted logistic regression and subgroup analyses are presented as odds ratios (ORs) with 95% confidence intervals (CIs). All statistical analyses were conducted using SPSS (Version 29), with a significance threshold set at *p* < 0.05 and 95% CI to denote statistical significance.

## 3. Results

After multiple screenings, 2543 cases were selected from the NHANES database (2017 to March 2020), representing a total population sample of 56,617,830. Figure 1 presents the case selection process in a flowchart. The weighted mean age of the hypertensive participants was 60.95 ± 13.15 years. The age distribution of the hypertensive participants was as follows: 13.7% were between 18 and 45 years old, 45.6% were between 46 and 65 years old, and 40.7% were 66 years old and older. Approximately 52% of the hypertensive participants were female, and around 64% were married or living with a partner. The racial composition was 67% non-Hispanic White, 13.5% non-Hispanic Black, 6.3% other Hispanic, 4.4% non-Hispanic Asian, and 4.3% multi-racial. Over 85% of the hypertensive participants were born in the United States, and about 42.5% had a high school education or less. Regarding medication use, the weighted mean number of medications used was 4.74 ± 3.17. About 27% of the hypertensive participants used 1 to 2 medications or 3 to 4 medications, while approximately 19% used 5 to 6 medications. Furthermore, 13%, 7%, and 6% of the hypertensive participants used 7 to 8 medications, 9 to 10 medications, and 11 or more medications, respectively (Table 1).

### 3.1. Prevenance of Depression Symptoms

The weighted mean PHQ-9 score was 3.81 ± 4.65, with approximately 12.3% of hypertensive participants meeting the criteria for depression (PHQ-9 score ≥ 10) (weighted *n* = 6,949,529). Hypertension patients aged 18 to 45 years exhibited a higher prevalence of depressive symptoms compared to those aged 46 to 65 years and 66 years and older (*p* = 0.035). Female patients with hypertension had a significantly higher rate of depressive symptoms than their male counterparts (*p* = 0.003). Individuals identified as multi-racial experienced a higher rate of depressive symptoms compared to those in other racial categories (*p* = 0.005). Patients with a high school education or less had a higher incidence of depressive symptoms than those with a college or graduate degree (*p* = 0.006). Additionally, hypertension patients taking 11 or more medications had the highest rate of depressive symptoms, while those using only 1 to 2 medications had the lowest rate (*p* < 0.001) (Table 2).

### 3.2. The Association Between Number of Medication Use and Depression Symptoms

Weighted multivariate binary logistic regression models were constructed to estimate the effect of the number of medications used on depressive symptoms among hypertensive patients, adjusting for other sociodemographic characteristics. The model was statistically significant, explaining approximately 18.3% of the variance in the dependent variable (Nagelkerke R^2^ = 0.183). Age, gender, marital status, education level, and the number of medications used were significantly associated with depressive symptoms in hypertensive patients.

Hypertensive patients aged 66 years and older were less likely to experience depressive symptoms compared to those aged 18 to 45 years (OR = 0.307, *p* = 0.004). However, female patients were more likely to experience depressive symptoms compared to male patients (OR = 1.47, *p* = 0.032). Moreover, those who were widowed, divorced, or separated were more likely to have depressive symptoms than those who were married or living with a partner (OR = 1.72, *p* = 0.032). Similarly, patients with hypertension who were never married were more likely to have depressive symptoms than those who were married or living with a partner (OR = 2.90, *p* < 0.001). Patients with a graduate degree were less likely to experience depressive symptoms compared to those with a high school diploma or less (OR = 0.46, *p* = 0.005).

Regarding the number of medications used, patients taking five to six medications were 4.5 times more likely to experience depressive symptoms compared to those taking one to two medications (OR = 4.51, *p* < 0.001). Similarly, patients using seven to eight medications were approximately 4.5 times more likely to have depressive symptoms compared to those using one to two medications (OR = 4.47, *p* < 0.001). Patients taking nine to ten medications were five times more likely to have depressive symptoms compared to those taking one to two medications (OR = 4.97, *p* < 0.001). Finally, patients using eleven or more medications were 10 times more likely to experience depressive symptoms compared to those using one to two medications (OR = 10.06, *p* < 0.001) (Table 3, Figure 2).

## 4. Discussion

The objective of this study was to assess the correlation between the number of medications and the rate of depressive symptoms with reference to the survey conducted between 2017 and 2020 under NHANES. Thus, the results of our study show that the prevalence of depressive symptoms is positively related to polypharmacy in hypertensive patients. According to the weighted data, the total sample of the hypertensive population for the whole United States is more than 56 million; therefore, the given health issue can be characterized as sufficiently covered.

The hypertensive population in our study had a mean age of approximately 61 years, with a nearly even gender distribution (52.4% female, 47.6% male). Notably, 85.6% of the participants were born in the U.S., and over 42% had a high school education or less, suggesting that lower educational attainment might correlate with increased health challenges. The average number of medications used by participants was 4.74, with 27.9% of participants taking 1–2 medications and another 27.5% taking 3–4. This distribution highlights the prevalence of polypharmacy within the hypertensive population, as nearly half of the participants took more than four medications.

Based on our study results, we see that 12.3 percent of hypertensive participants showed depressive symptoms with a PHQ-9 score of 10 and above. This prevalence is particularly higher amongst young people, females, the less educated, and those with higher drug consumption rates. In general, 16.1% of given hypertensive patients with depressive symptoms were found to be in the age range of 18–45 years, and this difference was statistically significant (*p* = 0.035). This trend is in agreement with the outcomes of other research which reveal that younger adults with chronic diseases such as hypertension are at a higher risk of developing depression [23]. Female respondents in our study were considerably more likely to experience depression than male respondents (15.1% vs. 9.2%, *p* = 0.003)—a finding that is consistent with prior research on gender disparities in mental health [24,25].

One of the most striking findings from our study is the strong association between the number of medications used and the prevalence of depressive symptoms. Hypertensive patients on 11 or more medicines were 10 times more likely to have depressive symptoms than those on 1–2 medicines (OR = 10.06, *p* < 0.001). This trend was seen irrespective of the degree of polypharmacy; patients who received 5–6 prescriptions had an approximately 4.5-fold increased risk of acquiring depression (OR = 4.51, *p* < 0.001). Patients using 9–10 medications had a nearly five-fold risk of developing depression (OR = 4.97, *p* < 0.001). These results are in line with other works that have also pointed out polypharmacy as an independent risk factor for depression particularly among the elderly [26].

There was a significant multifactorial interaction between sociodemographic characteristics medication use and depression. Hypertensive patients aged 66 years and above were less likely to report depressive symptoms compared to those aged 18–45 years (OR = 0.307, *p* = 0.004), which mean young adults are more distressed when it comes to managing multiple drugs. This is in disagreement with some studies that reveal higher rates of depression among older people, but it could be due to different ways of handling stress or different phases of life [27]. In addition, female hypertensive patients had a significantly higher probability of depressive symptoms compared to male patients (OR = 1.47, *p* = 0.032), which is in concordance with other studies also suggesting gender inequalities in mental health [28]. Marital status was also significant, with those who have never been married or widowed/divorced/separated being more likely to have depression than those who are married or in a common-law union (OR = 2.90, *p* < 0.001 and OR = 1.72, *p* = 0.032, respectively).

Moreover, patients with a graduate education degree were less likely express depression symptoms compared with those with high school or less education degree (OR = 0.46, *p* = 0.005). Educational attainment has been consistently shown to influence health outcomes, including mental health. Higher education levels are often associated with increased health literacy, better access to healthcare, and healthier lifestyle choices, which can help individuals manage chronic conditions such as hypertension more effectively, potentially reducing the risk of depression. Furthermore, individuals with a graduate degree are likely to experience fewer financial stressors and have access to better support systems, which may contribute to lower depressive symptoms. Several studies support this finding. For example, research has demonstrated that lower educational attainment is a risk factor for depressive symptoms, as individuals with less education may have limited coping resources and may be more likely to face socioeconomic hardships, which exacerbate mental health issues [29,30]. In contrast, higher educational attainment often correlates with increased problem-solving skills and emotional resilience, which can mitigate the psychological burden of chronic illnesses like hypertension [31].

The findings of our study support previous studies that found polypharmacy as one of the predictors of depression. In detail, Masnoon et al. [5] performed a systematic review in which they pointed out that multiple medication users of the elderly population are at a higher risk of adverse drug reactions, such as psychological disturbances, anxiety, and depression. This might be attributed to the accumulation of effects of the various medications, possible drug interactions, and the general implications of having to manage various treatment regimens [6]. Polypharmacy in this situation refers not only to the physical load, but can also be associated with psychological stress, particularly in cases where patients have to manage different drugs and their possible side effects [7].

These findings are consistent with our study because the hypertensive patients who reported to be using more than five medications were more likely to report depressive symptoms as compared to the patients who were using fewer medications or none at all. This strong association strengthens the belief that the complexity and burden of medications worsen with the risk of depression and accentuates polypharmacy as an important predictor of deteriorating mental health among hypertensive patients.

Furthermore, the study by Gottlieb et al. [32] also supports our results on the relationship between the risk of depression and the patient gender and age. It noted that females and young people with chronic diseases were found to be more sensitive to depression by reasons correlated with biological, psychological, and social aspects. Hormonal changes that are characteristic of the female sex may predispose them to depression, and young people may have increased emotional and social stress associated with their illness, which can cause feelings of frustration, isolation, or hopelessness [32].

In the context of our study, the higher prevalence of depressive symptoms in young patients with hypertension and females can be explained by similar factors. It may be challenging for the younger patient to effectively manage hypertension through polypharmacy and women may be more likely to exhibit an emotional response to the stress of having to deal with multiple medications and physicians.

Although other studies have proved that polypharmacy causes depression, the present study builds on those findings by investigating patients with hypertension, a group of people who are already vulnerable because of their illness. More notably, we adjusted for a host of other demographic factors including age, gender, marital status, education level, and race, which facilitated a better examination of the direct effects of medication use on depression status. Thus, the present study contributes to the knowledge of how polypharmacy impacts hypertensive patients by identifying issues that are specific to this patient population and can guide clinicians in designing effective interventions to address patients’ physical and psychological well-being.

In another research, the significance of exploring the relationship between polypharmacy practices and depressive symptoms in individuals managing chronic illnesses was underscored by FENG et al. [33]. Depressive symptoms were assessed, and through comprehensive statistical analyses, it was discovered that an increased likelihood of experiencing depressive symptoms was associated with patients engaged in polypharmacy, with an odds ratio of 1.58 (95% CI: 1.17–2.16) compared to those on fewer medications. It was identified that factors associated with a decreased risk of depressive symptoms included higher levels of educational attainment, regular physical exercise, and the use of Chinese medicine (CM) for treatment. The conclusion drawn by the researchers was that increased depressive symptoms were significantly associated with unnecessary multiple drug use in individuals with chronic diseases, and it was emphasized that integrating patients’ health information and educational background into strategies for managing depression in those with chronic conditions is of great importance [33].

It has to be noted that in individuals with long-standing conditions such as hypertension, the risk of developing behavioral disorders can be significantly increased by psychological distress, including anxiety and depression. A study conducted at two specialized hospitals in Ghana, involving a sample of 400 patients with hypertension, was aimed at investigating the prevalence of psychological issues such as tension and depression, as well as the overall impact of psychological distress on adherence to prescribed medication. This research was conducted to understand how these psychological factors influence the ability of patients to consistently follow their treatment regimens. It was indicated by the findings that tension (56%), distress (20%), and depression (4%) were experienced by a considerable number of patients [34].

This study of the correlation between polypharmacy and depression in hypertensive patients provides a deeper understanding of the necessity of using complex comprehensive care models to address the medication effects on the patient’s physical and mental health. It also backs deprescribing projects in which practitioners analyze the patient’s prescription list and rationalize the medication regimen to minimize the psychological load.

### 4.1. Implications for Clinical Practice

Regarding clinical practice, an important finding here is a strong relationship between the number of medications and depressive symptoms in hypertensive patients. It is crucial for healthcare providers to closely watch the patients on multiple medications since they are most vulnerable to experiencing a depressive episode. The need to incorporate routine mental health checks, especially for young people, women, and those who are on multiple drugs, cannot be overemphasized. Moreover, the use of deprescribing interventions may be required to reduce the potential polypharmacy and the adverse effects related to the use of multiple medications such as depression.

### 4.2. Limitations and Future Research

The following limitations should be considered despite this study providing valuable insight into the correlation between medication use and depression among hypertensive patients. First, while we investigated the effect of polypharmacy on depression, we did not differentiate between specific types of medications used by those on multiple drugs. This limits our ability to determine whether particular classes of medications might have a stronger association with depression than others. Second, depression was assessed using the self-reported PHQ-9 scale, which may be subject to reporting bias and misclassification. Although the PHQ-9 is a validated tool, its reliance on self-reporting might not capture all instances of depression with clinical accuracy. Third, the use of NHANES data presents limitations regarding the generalizability of our findings. Fourth, our study focused on the number of medications taken but did not account for underlying medical conditions other than hypertension that may influence both the use of multiple medications and the risk of depression. Fifth, this study did not exclude medications that are used to treat or may potentially cause depression or anxiety, which could influence the observed associations between medication use and depression. Sixth, the cross-sectional design of NHANES weakens our ability to determine causality between polypharmacy and depression. Seventh, the number of medications used was self-reported and may be subject to reporting bias. Lastly, future research should employ longitudinal designs to establish the direction of effects between polypharmacy and depression and to examine the efficacy of deprescribing in hypertensive patients.

## 5. Conclusions

To conclude, the findings of this study demonstrate that polypharmacy is positively correlated with depression among hypertensive patients according to NHANES 2017–2020 data. These results highlight the associations with the overall impact of polypharmacy on mental health in chronic illness care. Thus, the outcomes reveal that patients who use four or more medications daily are more likely to experience depression than those who use fewer medications. These results are consistent with prior research indicating that multiple prescription use might exacerbate mental health problems through various and possibly interdependent pharmaceutical impacts, medication compliance issues, and other relevant psychological factors.

Healthcare providers, especially those managing hypertensive patients, can improve treatment regimens based on the findings highlighted above. Interventions may include annual medication reconciliation, reduced medication use, and monitoring of mental health disorder risk among polypharmacy users. However, it is pertinent to point out some limitations of this research. Due to the cross-sectional nature of NHANES data, determining causality is challenging, and self-reported data may introduce bias. Future studies should employ longitudinal data to describe the long-term impact of polypharmacy on mental health outcomes more accurately and guide specific interventions.

## Figures and Tables

**Figure 1 medicina-60-01708-f001:**
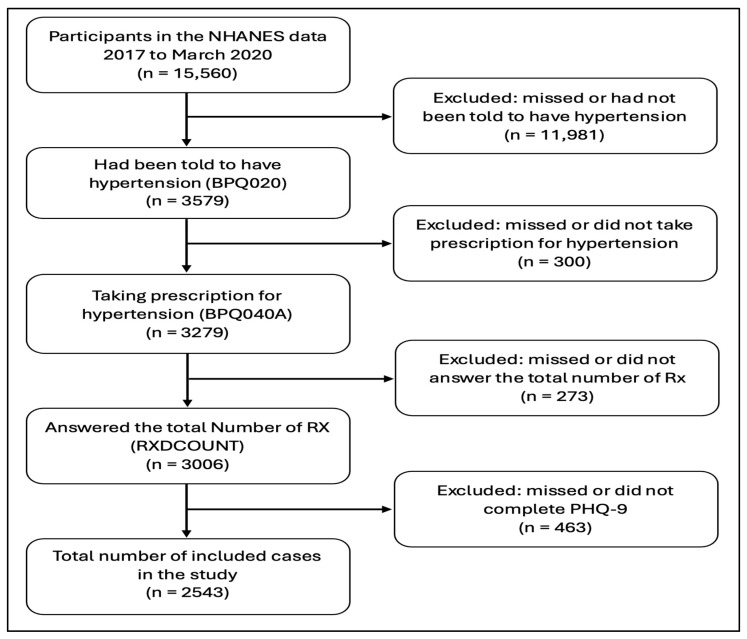
Cases section flowchart from the National Health and Nutrition Examination Survey (NHANES 2017–2020).

**Figure 2 medicina-60-01708-f002:**
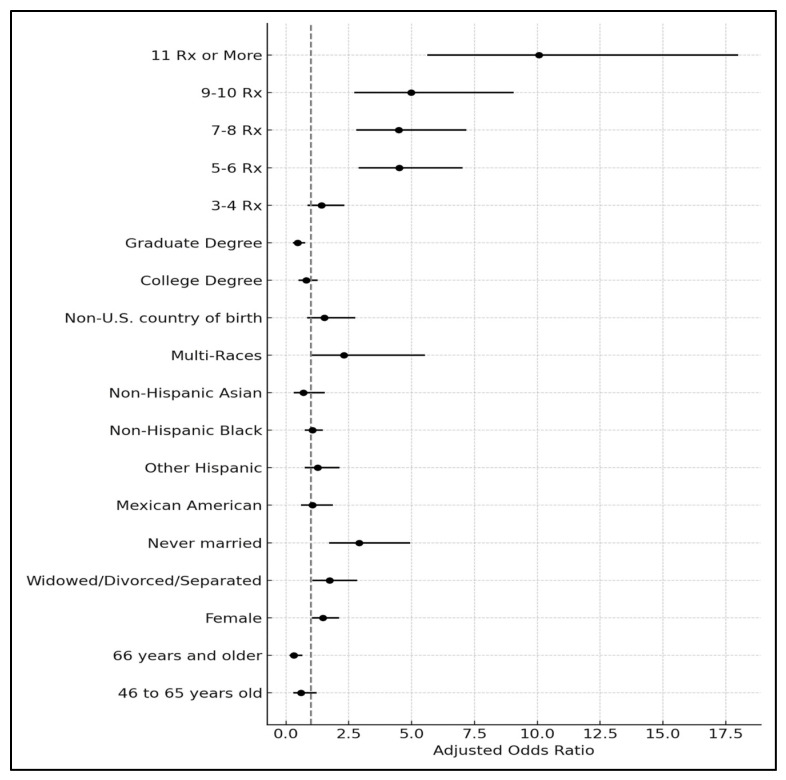
Adjusted odds ratio for depression across hypertensive patients with different numbers of medication use, adjusted for sociodemographic characteristics (weighted).

**Table 1 medicina-60-01708-t001:** Sociodemographic characteristics of selected cases (weighted *n* = 56,617,830).

Sociodemographic Characteristics	Frequency	Percent (%)
**Age (years)**	Mean: 60.95	SD: 13.15
18 to 45 years old	7,781,230	13.7%
46 to 65 years old	25,791,028	45.6%
66 years and older	23,045,571	40.7%
**Gender**		
Male	26,964,482	47.6%
Female	29,653,348	52.4%
**Marital Status**		
Married/Living with Partner	36,440,510	64.4%
Widowed/Divorced/Separated	15,804,524	27.9%
Never married	4,314,075	7.6%
Missing	58,720	0.1%
**Race**		
Non-Hispanic White	38,020,847	67.2%
Mexican American	2,471,818	4.4%
Other Hispanic	3,572,425	6.3%
Non-Hispanic Black	7,651,086	13.5%
Non-Hispanic Asian	2,475,948	4.4%
Multi-Races	2,425,706	4.3%
**Country of Birth**		
U.S. country of birth	48,441,833	85.6%
Non-U.S. country of birth	8,175,997	14.4%
Education Levels		
High School or Less	24,043,322	42.5%
College Degree	19,203,222	33.9%
Graduate Degree	13,311,346	23.5%
Missing	59,939	0.1%
**Number of Medication Use**	Mean: 4.74	SD: 3.17
1–2 Rx	15,804,679	27.9%
3–4 Rx	15,560,772	27.5%
5–6 Rx	10,669,663	18.8%
7–8 Rx	7,188,744	12.7%
9–10 Rx	4,086,023	7.2%
11 Rx or More	3,307,948	5.8%

**Table 2 medicina-60-01708-t002:** Characteristics of hypertensive patients included in the study stratified by depression conditions (weighted).

Sociodemographic Characteristics	Hypertensive Patients Without Depression (*n* = 49,668,301)	Hypertensive Patients with Depression (*n* = 6,949,529)	*p*-Value
**Age**			0.032
18 to 45 years old	13.1%	18.0%	
46 to 65 years old	44.7%	51.4%	
66 years and older	42.1%	30.6%	
**Gender**			0.003
Male	49.3%	35.7%	
Female	50.7%	64.3%	
**Marital Status**			<0.001
Married/Living with Partner	66.8%	47.7%	
Widowed/Divorced/Separated	26.7%	36.6%	
Never married	6.5%	15.7%	
**Race**			0.005
Non-Hispanic White	68.1%	60.4%	
Mexican American	4.3%	4.7%	
Other Hispanic	5.9%	9.0%	
Non-Hispanic Black	13.3%	14.7%	
Non-Hispanic Asian	4.7%	2.3%	
Multi-Race	3.6%	8.9%	
**Country of Birth**			0.356
U.S. country of birth	85.9%	83.2%	
Non-U.S. country of birth	14.1%	16.8%	
**Education Levels**			0.006
High School or Less	41.0%	53.2%	
College Degree	33.8%	34.8%	
Graduate Degree	25.1%	12.0%	
**Number of Medication Use**			<0.001
1–2 Rx	30.0%	12.7%	
3–4 Rx	29.1%	15.9%	
5–6 Rx	17.7%	26.7%	
7–8 Rx	11.9%	18.3%	
9–10 Rx	6.6%	11.4%	
11 Rx or More	4.6%	15.0%	

**Table 3 medicina-60-01708-t003:** Multivariate logistic regression for depression across hypertensive patients with different number of medication use, adjusted for sociodemographic characteristics (weighted).

Sociodemographic Characteristics	Adjusted Odds Ratio	95% CI	*p*-Value
**Age**			
18 to 45 years old	Ref.		
46 to 65 years old	0.598	(0.292–1.228)	0.154
66 years and older	0.307	(0.143–0.662)	0.004
**Gender**			
Male	Ref.		
Female	1.478	(1.038–2.105)	0.032
**Marital Status**			
Married/Living with Partner	Ref.		
Widowed/Divorced/Separated	1.727	(1.053–2.832)	0.032
Never Married	2.905	(1.708–4.941)	<0.001
**Race**			
Non-Hispanic White	Ref.		
Mexican American	1.048	(0.587–1.872)	0.869
Other Hispanic	1.260	(0.745–2.130)	0.373
Non-Hispanic Black	1.042	(0.743–1.461)	0.804
Non-Hispanic Asian	0.688	(0.305–1.554)	0.354
Multi-Race	2.309	(0.965–5.529)	0.059
**Country of Birth**			
U.S. country of birth	Ref.		
Non-U.S. country of birth	1.523	(0.843–2.751)	0.156
**Education Levels**			
High School or Less	Ref.		
College Degree	0.798	(0.503–1.266)	0.323
Graduate Degree	0.463	(0.278–0.772)	0.005
**Number of Medication Use**			
1–2 Rx	Ref.		
3–4 Rx	1.410	(0.856–2.322)	0.169
5–6 Rx	4.510	(2.897–7.023)	<0.001
7–8 Rx	4.479	(2.790–7.188)	<0.001
9–10 Rx	4.971	(2.730–9.053)	<0.001
11 Rx or More	10.066	(5.630–17.997)	<0.001

Ref. reference.

## Data Availability

The NHANES data are publicly available on the CDC website: https://www.cdc.gov/nchs/nhanes/index.htm (accessed on 1 January 2024).

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
