# Peer review of "Investigating the Impact of the Number of Medication Use on Depression Among Hypertensive Patients: Results from the National Health and Nutrition Examination Survey Database"

_medicina, 2024, doi:10.3390/medicina60101708_

Round 1

Reviewer 1 Report

Comments and Suggestions for Authors

At the beginning, I would like to congratulate the authors of the paper. The impact of polypharmacy on the mental health of patients is an important topic that needs to be explored more closely. I have a few questions and comments.

At the beginning of the paper (methods), it is not specified precisely whether the number of drugs taken by patients refers only to hypertension drugs. My guess is that it does, however, I obtained this information only in the discussion (line 246). This information should have been given at the beginning.

Current studies indicate that not only polypharmacy has an impact on the patient's mental state but also the duration of treatment. Are the authors able to determine how long the treatment lasted in the study group? That is, for how many years had the patients been taking medication for hypertension?

Is it possible that the respondents gave the total number of medications they are taking, including all the medications they are taking for potentially other comorbidities?

In section 3.2 Prevention of depression symptoms, the authors repeat the phrase “Hypertensive participants” several times. I suggest recasting the sentences to use synonyms or starting the sentence simply by describing the results.

Similarly, in section “3.3 The association between number of medication use and depression symptoms” the authors repeat the phrase “Hypertensive partients” several times. Repeating the same phrases over and over again, in the same chapter, reduces the quality of this paper.

Hypertensive patients with a graduate degree were less likely to experience depressive symptoms compared to those with a high school diploma or less (OR = 0.46, p = 228 0.005)."-Why? What do other researchers show on this topic? Please expand the discussion.

Author Response

Please look to the attached file.
Thank you!

Reviewer 2 Report

Comments and Suggestions for Authors

Dear authors,

I think the idea of studying polypharmacy as a possible causal agent for depression and anxiety seems interesting to explore. However, I have few methodological concerns:

1-Since the authors have not assigned the exposure to the participants, we are dealing with an observational study. Observational studies require to be reported using the STROBE checklist.

2-I strongly recommend to read the documentation regarding the exposure variable at https://wwwn.cdc.gov/Nchs/Nhanes/2017-2018/P_RXQ_RX.htm. I think several variables allow to identify drugs and conditions that might impair the analyses and try to recount the number of drugs due to the following concerns: 

 2-a-Some prescription drugs are used for treating depression and anxiety should be excluded from the count, due to clear reverse causality.

 2-b-Drugs which are well known to cause anxiety and/or depression as adverse effects should also be excluded from the count.

3-I strongly recommend not to present Table 2 with row percentages, do it by columns.

4-Have the authors considered using causal directed acyclic diagram (DAGs) for adjusting the logistic regression models?

Author Response

(The authors gave the same response as above.)
